# Analysis of Cholera Risk in India: Insights from 2017–18 Serosurvey Data Integrated with Epidemiologic data and Societal Determinants from 2015–2019

Suman Kanungo[1], Ranjan Kumar Nandy[1], Rounik Talukdar[1], Manoj Murhekar[2], Andrew S. Azman[3,4,5], Sonia T. Hegde[3], Pranab Chatterjee[1,3], Debjit Chakraborty[1], Madhuchhanda Das[6], P Kamaraj[2], Muthusamy Santhosh Kumar[2], Dengue Serosurvey Group[¶], Shanta Dutta[1]*

1 Indian Council of Medical Research–National Institute of Cholera and Enteric Diseases, Kolkata, India, 2 Indian Council of Medical Research–National Institute of Epidemiology, Chennai, India, 3 Johns Hopkins University Bloomberg School of Public Health, Baltimore, Maryland, United States of America, 4 Geneva Centre for Emerging Viral Diseases, Geneva University Hospitals, Geneva, Switzerland, 5 Division of Tropical and Humanitarian Medicine, Geneva University Hospitals, Geneva, Switzerland, 6 Indian Council of Medical Research, New Delhi, India

¶ Membership of Dengue Serosurvey Group is provided in the Acknowledgements.
* shanta1232001@gmail.com, shanta.niced@icmr.gov.in, drshantadutta@gmail.com

**Data Availability Statement:** Data regarding the cholera surveillance were acquired from public portal of IDSP, which although, is available as an

## Abstract

### Background

Indian subcontinent being an important region in the fight to eliminate cholera needs better cholera surveillance. Current methods miss most infections, skewing disease burden estimates. Triangulating serosurvey data, clinical cases, and risk factors could reveal India's true cholera risk.

### Methods

We synthesized data from a nationally representative serosurvey, outbreak reports and risk factors like water, sanitation and the Multidimensional Poverty Index, to create a composite vulnerability index for assessing state-wise cholera risk in India. We tested 7,882 stored sera samples collected during 2017–18 from individuals aged 9–45 years, for vibriocidal antibodies to *Vibrio cholerae* O1 using a cut-off titre ≥320 defining as elevated titre. We also extracted data from the 2015–19 Integrated Disease Surveillance Programme and published cholera reports.

### Results

Overall, 11.7% (CI: 10.4–13.3%) of the sampled population had an elevated titre of cholera vibriocidal antibodies (≥320). The Southern region experienced the highest incidence (16.8%, CI: 12.1–22.8), followed by the West (13.2%, CI: 10.0–17.3) and North (10.7%, CI: 9.3–12.3). Proportion of samples with an elevated vibriocidal titre (≥320) was significantly higher among individuals aged 18–45 years (13.0% CI: 11.2–15.1) compared to children

open-source information platform for any user, but imparts restriction on usage and duplication without necessary permissions from appropriate authorities. Data access for the serosurvey can be discussed via approaching through the following institutional email - director-niced@icmr.gov.in.

**Funding:** This work was funded with an intramural research grant to SD awarded by Indian Council of Medical Research, New Delhi bearing grant number 5/8-I (6)/2019-20-ECD-II dated 27/10/ 2020. The funder of the study had no role in study design, data collection, data analysis, data interpretation, or writing of the report.

**Competing interests:** The authors have declared that no competing interests exist.

9–17 years (8.6%, CI 7.3–10.0, p<0.05); we found no differences between sex or urbanicity. Between 2015–2019, the Integrated Disease Surveillance Program (IDSP) reported 29,400 cases of cholera across the country. Using the composite vulnerability index, we found Karnataka, Madhya Pradesh, and West Bengal were the most vulnerable states in India in terms of risk of cholera.

## Conclusion

The present study showed that cholera infection is present in all five regions across India. The states with high cholera vulnerability could be prioritized for targeted prevention interventions.

## Author summary

Cholera remains a significant public health threat in areas with inadequate water, sanitation, and hygiene (WASH). The World Health Organization's Global Task Force on Cholera Control (GTFCC) launched "Ending Cholera: A Global Roadmap to 2030" to reduce cholera deaths and eliminate the disease in 20 countries by 2030 through targeted interventions, including improved WASH practices and disease surveillance. cholera surveillance, especially in developing countries seldom poses difficulty in terms of the need for laboratory infrastructure, and other resource constraints. Thereby, in this study, we utilized a viable approach with serosurvey data from 2017, cholera outbreak data, WASH indicators, and the Multidimensional Poverty Index (MPI) to create a composite vulnerability index for cholera risk in India. Overall, 11.7% of the sampled population had detectable levels of cholera antibodies reflecting previous infection. The Southern region of India experienced the highest incidence, followed by the West and North. the study showed that cholera infection is present in all five regions across India. Further, the vulnerability index which we built could be utilized in other settings so that effective tailored interventions could be applied in identified priority areas.

## Introduction

Cholera continues to be a major public health threat in communities lacking safe drinking water, sanitation, and hygiene (WASH) [1]. Although low- and middle-income countries have made numerous concerted efforts to improve WASH practices to contain cholera, lack of precise reporting and surveillance has continued to provide inaccurate burden estimates [2–4]. In India, since 2004, the Integrated Disease Surveillance Program (IDSP) has been conducting surveillance and response activities for outbreak-prone diseases, including cholera [5,6]. Despite years of cholera research, the burden and geographic variability of the disease across India are poorly understood due to limited laboratory capacity, overreliance on outbreak-based reporting, non-specific case definitions, and underreporting of cases due to apprehensions of negative impacts on trade, tourism and commerce [6,7]. A recent study analyzing disease surveillance data from India revealed a steady increase in reported cholera outbreaks across the country. While total of 68 cholera outbreaks were reported from 1997 to 2006, this number rose to 559 during 2009 to 2017 [4,8].

In 2017, the World Health Organization Global Task Force on Cholera Control (GTFCC) endorsed a new strategy "Ending Cholera: A Global Roadmap to 2030". The goal of this strategy is to achieve a 90% reduction in deaths and elimination in 20 countries by 2030 through targeted multisectoral interventions, including disease surveillance to identify cholera hotspots, introduction of oral cholera vaccine, enhancement in WASH practices, and improvement in case management [9,10].

Identifying priority areas for interventions is key for the roadmap to be effective. Outbreak reports, routine acute watery diarrhoea surveillance data, serosurvey data and the prevalence of known cholera risk factors all provide partial understanding of the distribution of the disease risk. Triangulating these data sources together can help provide a more comprehensive picture of cholera risk and aid in the prioritization of cholera control resources [4]. Here, we combined data from a nationally representative serosurvey with data on clinical cholera cases and risk factors to provide new insights into the distribution of cholera in India.

## Methods

### Ethics statement

This study had been approved by the institutional ethics committee of Indian Council of Medical Research–National Institute of Cholera & Enteric Diseases, Kolkata, India (ICMR–NICED) bearing id. A-1/2020-IEC dated 5[th] February 2020. The sera samples utilized in this study was collected and preserved during national dengue serosurvey, conducted by ICMR National Institute of Epidemiology (ICMR-NIE), Chennai, India from June 2017 to April 2018 [11]. The dengue serosurvey study protocol was approved by the Institutional Ethics Committees of ICMR- NIE and all participating institutes. Written informed consent was obtained for the use of sera samples in the dengue study and their subsequent use in anonymized form for further research and scientific development. Written informed consent from people aged 18 years and older, parental consent from parents of children aged between 5–17 years, and assent from children aged between 7–17 years was obtained [11].

### Study design

We generated vibriocidal data using serum samples (n = 7882) collected for the national dengue serosurvey, conducted by ICMR National Institute of Epidemiology (ICMR-NIE), Chennai, India from June 2017 to April 2018 [11]. For the 15 states where the serosurvey was conducted, we then combined the state-specific vibriocidal data with the available reported data on cholera outbreaks from IDSP, WASH and the Multidimensional Poverty Index (MPI) from the National Family Health Survey-4 to develop a composite index for comparing cholera risk nationally [12,13].

### Cholera serosurvey

**Sampling strategy.** The sampling strategy has been described elsewhere in detail [11]. Briefly, the serosurvey was conducted in five geographic regions of India (i.e., North, Northeast, East, West and South) and targeted three age groups (i.e., 5–8, 9–17 and 18–45 years). From each region, three states were selected randomly (total 15 states), and from each state, four districts were selected by probability proportional to population size. From each district, four clusters (2 in urban and 2 rural areas) were selected randomly. One census enumeration block (CEB) was selected randomly from each cluster. In India, during decennial census operations, an enumerator is allotted to one CEB, which has about 120–150 households. The selected CEB was then enumerated, and 25 individuals were randomly selected per age group

(5–8 years, 9–17 years, 18–45 years). Anonymised sera samples from a total of 12,300 individuals from 15 states was included into this study (4265 children (9–17 years) and 3976 adults (18–45 years)). Each administrative zones of India, and the study locations has been depicted in S1 Fig (The base layer of the map utilized to create this map is taken from open-source platform https://www.indianremotesensing.com/2017/01/Download-India-shapefile-with-kashmir.html).

### Inclusion of samples in current study

In this study, we estimated the extent of cholera infection among individuals aged 9 to 45 years corresponding to the year 2016. The samples were collected in 2017 and at a vibriocidal titre of 1: 320, infection occurring in past 365 days could be measured. Viability of stored sera samples was maintained by archiving samples at -80˚C. Frozen aliquots of samples were transported to ICMR-National Institute of Cholera and Enteric Diseases, Kolkata, from ICMR-NIE, Chennai under cold chain, maintaining the required temperature. Received sera aliquots were stored at -80˚C till accessed for vibriocidal assays. Sera from children aged 5–8 years could not be analysed due to inadequate (less than 60 µl) sample volume.

### Laboratory procedures–vibriocidal assay

To eliminate endogenous complement activity, frozen sera samples were thawed from -80˚C and heat-inactivated at 56˚C for 30 minutes. Using 25 µl of sterile normal saline as diluent and a starting dilution of 1:10, each serum was serially half-diluted till the 12$^{th}$ well of a sterile micro-dilution plate. 25 µl of bacteria-complement mixture (final dilution of complement 1:140) was added to each well of a micro-dilution plate, mixed well, and incubated for 1 hour at 37˚C under mild shaking. Following incubation, each well was filled with 150 µl of BHI broth to stop the reaction. Plates were mixed and incubated at 37˚C for 3 hours. Optical density was then read at $OD_{595}$ on an ELISA (Enzyme-linked Immunosorbent Assay) plate reader [14,15].

Each plate contained a reference monoclonal antibody with a known reciprocal titre of 2560 against both the Inaba and Ogawa serotypes, and four wells each for growth controls and negative controls. Full growth of *V. cholerae* indicator strain (either Ogawa or Inaba) was determined by subtracting average values obtained with negative control wells (no organism) from average values of growth control wells (no test serum). The vibriocidal titer against Inaba and Ogawa cholerae serotypes was determined using indicator *V. cholerae* strains T-19479 and X-25049, respectively. Vibriocidal titre was determined as reciprocal of the highest dilution of test serum for which $OD_{595}$ reading was less than or same to the 50% of $OD_{595}$ obtained with background (negative control) subtracted growth controls. When 50% killing could not be achieved for a test serum, a titre value of 5 was assigned to indicate lower limit of detection [16].

### Cholera serosurvey

We defined elevated vibriocidal titre as the proportion of the sampled population that had a vibriocidal reciprocal titer ≥1:320. This titre was found to have a sensitivity of 80.6% and specificity of 83% in identifying individuals who had confirmed symptomatic *Vibrio cholerae* O1 infections within the previous year from blood collection in Bangladesh [2,11]. We estimated the elevated vibriocidal titre of cholera for each geographical region and each state along with 95% confidence intervals (CI), though we did not adjust for the performance of this threshold. To create a study map we used graduated colours based on the elevated vibriocidal titres (≥ 320) across 15 states of India instead of a zone-wise division. A classification matrix of

proportion of samples with elevated titres up to 5%, 5–10, 10–15 & 15–20 was used to achieve this. The locations of the districts where serosurvey was conducted along with the unweighted sero-incidence of cholera were integrated into Geographic Information System (GIS), and ArcGIS v. 10 (ESRI, Redlands, CA) was used for mapping [17]. The original survey as designed to estimate seroprevalence at the national and regional level, however, we incorporate state-level estimates from the serosurvey data into the cholera risk score, as one of the many components.

## Sensitivity analysis

In sensitivity analyses, we explored the use of a higher vibriocidal cut off titre, raising it from 320 to 640. This threshold has been shown to have a specificity of 91.4% and a sensitivity of 76.4% in identifying symptomatic infections in the last 100 days.

## Cholera outbreak surveillance

We reviewed weekly reports of cholera outbreaks from the Integrated Disease Surveillance Program (IDSP) reported during 2015–19 from the 15 Indian States where the serosurvey was conducted. We abstracted the number of annual outbreaks and cumulative cholera cases reported during 2015–19. Additionally, we conducted a literature search using PubMed and Google Scholar with the keywords "India", "cholera", "acute gastroenteritis", and "acute watery diarrhoea (or diarrhoea)". Publications were included if cholera cases were detected during 2015–19, indicating specific regions in India where the cases occurred, and specific dates of occurrence. Duplicate reporting of data compiled from various sources were addressed considering time and place of occurrence of the outbreak. A case of cholera was considered when *V. cholerae* O1 or O139 was isolated from any patient with diarrhoea [18]. Papers not written in English were excluded. Studies encompassing several years, but without a yearly breakdown for cholera cases, were also excluded from the review to have more granularity of the data. We also examined the annual reports published by ICMR institutes and reports of investigation of cholera outbreaks from grey literature.

## Data Triangulation and composite vulnerability index (CVI) for cholera

We combined the average number of cholera outbreaks per year reported during 2015–2019 with one-year serosurvey results as serological data reflects prior infection over a time period. Though vulnerability to cholera infection (both clinical and subclinical) and vulnerability to cholera disease (clinical) are not the same, literature suggests 80% of cholera infections are asymptomatic, and measuring both helps identify the true geographic distribution of disease [7,19]. We considered the following broad dimensions to create the composite vulnerability index: (1) cholera infections (from serosurvey and IDSP/literature search), (2) WASH data and (3) MPI [20]. For estimating cholera infection, we considered three parameters–elevated vibriocidal titres ($\geq$ 320), proportion of total state population living in cholera-affected districts (2015–2019), and average annual number of reported diarrhoea outbreaks where either cholera was confirmed or no other pathogen was detected (2015–2019) multiplied by number of outbreak years. For WASH indicators, we abstracted the following information from the 4[th] National Family Health Survey conducted during 2015–16 for the 15 States where the serosurvey was conducted: (a) proportion of households without improved water supply (b) proportion of households without improved sanitation. MPI examines areas such as education, health, and water sanitation conditions, among others, to identify not just the most vulnerable groups, but also to build a picture of where they fall short to inform potential policy interventions [21]. The description of each indicator along with their sources and steps of calculating

| | Description of the Indicators | Data Source | Example |
|---|---|---|---|
| $X_1$ | Recent Annual Infection (%) | Seropositivity estimated in current study | Seropositivity for Andhra Pradesh is 10.9%, so the score will be 10.9 |
| $X_2$ | Proportion (%) of population of cholera affected districts (2015-2019) out of total population of state | IDSP weekly reporting data, Census 2011 | Out of 13 districts in Andhra Pradesh, only 1 district is cholera affected. The population of that district contributes to 4.8% out of total population of the state. Hence the score is 4.8 |
| $X_3$ | Average annual number of reported diarrhoea outbreak (2015-2019) multiplied by number of outbreak years | IDSP weekly reporting data | In Andhra Pradesh total number of Cholera outbreak reported in 5 years is 1 (only 1 year). Hence average number of cholera outbreak in a year is 0.2. Hence the score is 0.2* 1 year = 0.2 |
| $X_4$ | Proportion (%) of Household without improved sanitation | NFHS-4 | For Andhra Pradesh 46.4 % of Household do not have improved sanitation. Hence the score is 46.4 |
| $X_5$ | Proportion (%) of Household without improved drinking water | NFHS-4 | For Andhra Pradesh 27.3 % of Household do not have improved sanitation. Hence the score is 27.3 |
| $X_6$ | Multidimensional Poverty Index/ MPI (converted in 100-point scale). | NITI Aayog -Baseline Report, Based on NFHS-4 (2015-16) | MPI for Andhra Pradesh is 0.053. In 100-point scale score is 6.5 |
| *So, For Andhra Pradesh The Composite Index: $X_1 + X_2 + X_3 + X_4 + X_5 + X_6 = 96.1$* | | | |

**Fig 1. Calculation of Composite Index used in the study.**

the composite vulnerability index (CVI) has been described in Fig 1 & S2 Table. Higher composite score signifies more vulnerability. The score only gives a relative ranking of 15 states.

Each of the six indicators was given equal weight (1/6) in our CVI calculation. Although, MPI includes sanitation and drinking water quality parameters (both were a weight of 1/21 or, 4.76% each), we included WASH data as a separate parameter (X4, X5: Fig 1) in our CVI to increase their weight to ~17.4% each. Sustainable WASH solutions for populations most at risk have been highlighted as one of the most crucial measures to control cholera [12]. (MPI chiefly includes three dimensions with 1/3 weight given to each, further the "standard of living" dimension has seven indicators, thereby making a weight of 1/21 for each of its indicator) [22]. To see the composite vulnerability ranking across each state as compared to ranking based on each individual indicators (WASH/ MPI, outbreak trend wise, and proportion of sampled population with elevated vibriocidal titre $\geq$ 320), we created a heat map. In this heat map, the more the scores more the condition being assessed was deemed as worse. (i.e., the more the composite score the more vulnerability to cholera).

## Results

### Serosurvey

We tested a total of 7882 sera samples from five geographic regions; about 52% (n = 4104) sera were from children aged 9–17 years and the remaining 3778 (47.9%) were from individuals aged 18–45 years. All the cases identified were of vibrio cholera O1, and no case of vibrio cholera O139 was found. The overall proportion of sampled population with an elevated vibriocidal titre $\geq$ 320, of cholera infection was 11.7% (95% CI: 10.4% - 13.3%). The highest estimate was in the Southern (16.8%, 95% CI: 12.1–22.8) region, followed by the Western (13.2%, 95% CI: 10.0–17.3) and Northern region (10.7%, 95% CI: 9.3–12.3), and lowest in the north-eastern

**Table 1. Estimation of proportion of sampled population with an elevated vibriocidal titre ($\geq$ 320), of V. cholera O1 using optimized vibriocidal reciprocal titer of 320 in different geographic regions of India, by selected socio-demographic characteristics during previous one year of serum collection (n = 7882).**

| Region | North | | North East | | East | | West | | South | | All Region | |
|---|---|---|---|---|---|---|---|---|---|---|---|---|
| Characteristics | Number Tested | Incidence (95% CI) | Number Tested | Incidence (95% CI) | Number Tested | Incidence (95% CI) | Number Tested | Incidence (95% CI) | Number Tested | Incidence (95% CI) | Number Tested | Incidence (95% CI) |
| **Age Group** | | | | | | | | | | | | |
| 9–17 Years | 819 | 9.3 (7.6–11.4) | 750 | 8.6 (4.0–17.5) | 852 | 3.8 (2.5–5.8) | 759 | 12.4 (9.8–15.6) | 924 | 10.1 (6.4–15.5) | 4104 | 8.6 (7.3–10.0) |
| 18–45 Years | 744 | 11.4 (9.5–13.6) | 807 | 7.0 (5.3–9.1) | 729 | 12.5 (9.0–17.1) | 681 | 13.5 (9.2–19.4) | 817 | 18.8 (13.0–26.5) | 3778 | 13.0 (11.2–15.1) |
| Overall | 1563 | 10.7 (9.3–12.3) | 1557 | 7.4 (5.4–10.1) | 1581 | 10.0 (7.3–13.4) | 1440 | 13.2 (10.0–17.3) | 1741 | 16.8 (12.1–22.8) | 7882 | 11.7 (10.4–13.3) |
| **Sex (9–17 Years)** | | | | | | | | | | | | |
| Male | 430 | 2.9 (1.3–6.4) | 359 | 9.2 (2.7–27.3) | 419 | 3.3 (1.4–7.6) | 379 | 14.1 (9.8–19.8) | 448 | 7.0 (3.6–13.2) | 2035 | 6.2 (4.4–8.6) |
| Female | 389 | 15.4 (12.3–19.1) | 391 | 8.1 (5.4–11.8) | 433 | 4.4 (2.5–7.5) | 380 | 10.9 (8.1–14.6) | 476 | 13.1 (8.7–19.4) | 2069 | 11.0 (9.0–13.4) |
| **Sex (18–45 Years)** | | | | | | | | | | | | |
| Male | 291 | 11.8 (6.4–20.6) | 269 | 7.8 (3.6–16.0) | 273 | 9.4 (5.4–15.8) | 285 | 12.6 (7.8–19.7) | 325 | 14.3 (10.1–20.0) | 1443 | 11.6 (9.0–14.8) |
| Female | 453 | 11.2 (7.5–16.4) | 538 | 6.6 (5.2–8.3) | 456 | 14.2 (10.3–19.3) | 396 | 14.3 (9.5–20.9) | 492 | 21.9 (14.5–31.7) | 2335 | 13.9 (11.5–16.7) |
| **Area of residence (9–17 Years)** | | | | | | | | | | | | |
| Rural | 393 | 8.7 (6.6–11.4) | 382 | 9.0 (4.3.18.0) | 439 | 3.3 (2.0–5.4) | 405 | 12.0 (9.0–15.8) | 473 | 8.4 (6.2–11.2) | 2092 | 7.8 (6.4–9.5) |
| Urban | 426 | 10.9 (7.5–15.5) | 368 | 2.4 (0.5–10.1) | 413 | 8.6 (5.3–13.6) | 354 | 14.6 (10.9–19.3) | 451 | 13.8 (5.4–31.0) | 2012 | 11.6 (8.7–15.3) |
| **Area of residence (18–45 Years)** | | | | | | | | | | | | |
| Rural | 332 | 11.0 (8.8–13.7) | 404 | 6.7 (5.0–9.1) | 377 | 11.8 (8.0–17.0) | 343 | 14.1 (8.9–21.5) | 425 | 17.6 (10.4–28.2) | 1881 | 12.5 (10.3–15.0) |
| Urban | 412 | 12.1 (8.8–16.4) | 403 | 9.5 (7.2–12.4) | 352 | 18.5 (12.8–25.9) | 338 | 11.4 (7.8–16.5) | 392 | 21.6 (14.0–31.8) | 1897 | 14.9 (11.9–18.6) |

region (7.4%, 95% CI: 5.4–10.1) (Table 1). Elevated vibriocidal titres ($\geq$ 320) was significantly noted among individuals aged 18–45 years (13.0% 95% CI: 11.2–15.1) as compared to children aged 9–17 years (8.6%, 95% CI 7.3–10.0, p<0.05). This did not differ significantly by sex and area of residence (i.e., urban vs. rural).

The un-weighted proportion of individuals with vibriocidal cut-off reciprocal titre of > = 320 was highest in Tamil Nadu 16.7%, (93/556); followed by Madhya Pradesh 13.4%, (70/524); Uttar Pradesh 12.6%, (69/548); Rajasthan 12.6%, (62/492); and West Bengal 12.1% (58/479). The proportion was lowest in the North-eastern region, specifically in Meghalaya 4.4%, (23/519) and Assam 5.1%, (26/507). (Table 2). Cholera serosurvey data showed clusters of cholera susceptibility alongside the heterogenous distribution of cases in the country. District wise serosurvey results across the 15 sampled states is given in S1 Table and S2 Fig.

Overall, 5.2% (95% CI 4.2–6.2) of the sampled population had a vibriocidal titre $\geq$640 (S5 Table), similar to the primary serosurvey results, with highest estimates coming from the Southern (8.4%, CI: 5.6–12.5) and Western (5.1%, CI: 3.6–7.3) regions.

## Cholera Outbreaks Identified in India from Literature Survey (2015–2019)

The literature survey revealed that 29,400 cases were reported from different states and Union Territories (UTs) during 2015–19 (Figs 2 and 3). 22 (61.1%) of 36 States/UTs and 124 (19.3%)

**Table 2. Incidence of Cholera using vibriocidal reciprocal cut off titre of 320 in different states of India during past one year of serum collection.**

| State Name | Number of sera tested / Number of sera with reciprocal vibriocidal titer ≥ 320 (n/N) | Elevated vibriocidal titre (≥ 320) | No. of districts with Cholera / Total surveyed districts (n / N) | No. of years affected | Total no. of cases |
|---|---|---|---|---|---|
| Tamil Nādu | 93 / 556 | 16.7 | 2 / 32 | 2 | 6 |
| Madhya Pradesh | 70 / 524 | 13.4 | 14 / 50 | 4 | 4040 |
| Uttar Pradesh | 69 / 548 | 12.6 | 1 / 71 | 2 | 333 |
| Rajasthan | 62 / 492 | 12.6 | 2 / 33 | 5 | 1026 |
| West Bengal | 58 / 479 | 12.1 | 11 / 19 | 4 | 5291 |
| Karnataka | 61 / 531 | 11.5 | 17 / 30 | 5 | 1277 |
| Maharashtra | 48 / 424 | 11.3 | 16 / 35 | 5 | 8040 |
| Andhra Pradesh | 71 / 654 | 10.9 | 1 / 13 | 1 | 22 |
| NCT of Delhi | 49 / 501 | 9.8 | 3 / 9 | 3 | 187 |
| Bihar | 46 / 530 | 8.7 | 0 / 38 | 0 | 0 |
| Punjab | 43 / 514 | 8.4 | 8 / 20 | 5 | 4561 |
| Tripura | 38 / 531 | 7.2 | 0 / 4 | 0 | 0 |
| Odisha | 30 / 572 | 5.2 | 10 / 30 | 5 | 541 |
| Assam | 26 / 507 | 5.1 | 6 / 27 | 4 | 661 |
| Meghalaya | 23 / 519 | 4.4 | 0 / 7 | 0 | 0 |

of the 641 districts reported cholera during the study period. During this period, 10 states reported cholera outbreaks three or more times among the years studied (S3 Table). Six states (Gujarat, Karnataka, Maharashtra, Odisha, Punjab, Rajasthan) reported cholera cases every year for the 5 years studied. Among the districts, Gandhinagar, Anand and Vadodara in Gujarat, Nashik in Maharashtra, Ludhiana and Hoshiarpur in Punjab, Purulia in West Bengal, Chandigarh and Dadra and Nagar Haveli reported outbreaks in 3 or more years (S3 Table). A total of 268 diarrheal outbreaks were reported to the IDSP, five states (West Bengal, Karnataka, Punjab, Madhya Pradesh, and Gujarat) accounted for 60% of these outbreaks.

### Composite vulnerability index

We generated composite vulnerability scores for the 15 states in the current study. Karnataka had the highest with a vulnerability score of 170.8, followed by Madhya Pradesh (169.4), West Bengal (168.8), Orissa (161.1), and Maharashtra (138.7). Tripura recorded the lowest score of 67.2; Tripura had no reported cholera cases from 2015 to 2019 (Table 3). States like Madhya Pradesh, Rajasthan, West Bengal, Karnataka, and Maharashtra had both higher proportions of sampled population with an elevated vibriocidal titre and vulnerability score. Complete description of the composite Index along with the indicators for individual states has been given in S2 Table. We, further produced a heat map where the CVI against individual WASH/ MPI, serosurvey data, and outbreak trend wise rank across the 15 states could be compared visually (**Fig 4**). The figure depicts that serosurvey data or any single variable did not always influence the composite score. The more vulnerable states like Karnataka, Madhya Pradesh, Odisha, West Bengal consistently ranked higher across all the indicators. However, states like Tamil Nadu despite having higher proportions of sampled population with an elevated vibriocidal titre, had a lower vulnerability rank due to improved WASH/ MPI and less annual outbreaks.

### Discussion

In this unique analysis, we have triangulated different data sources, incorporating cholera serosurvey data with cholera outbreak data from IDSP, WASH data from National Family Health

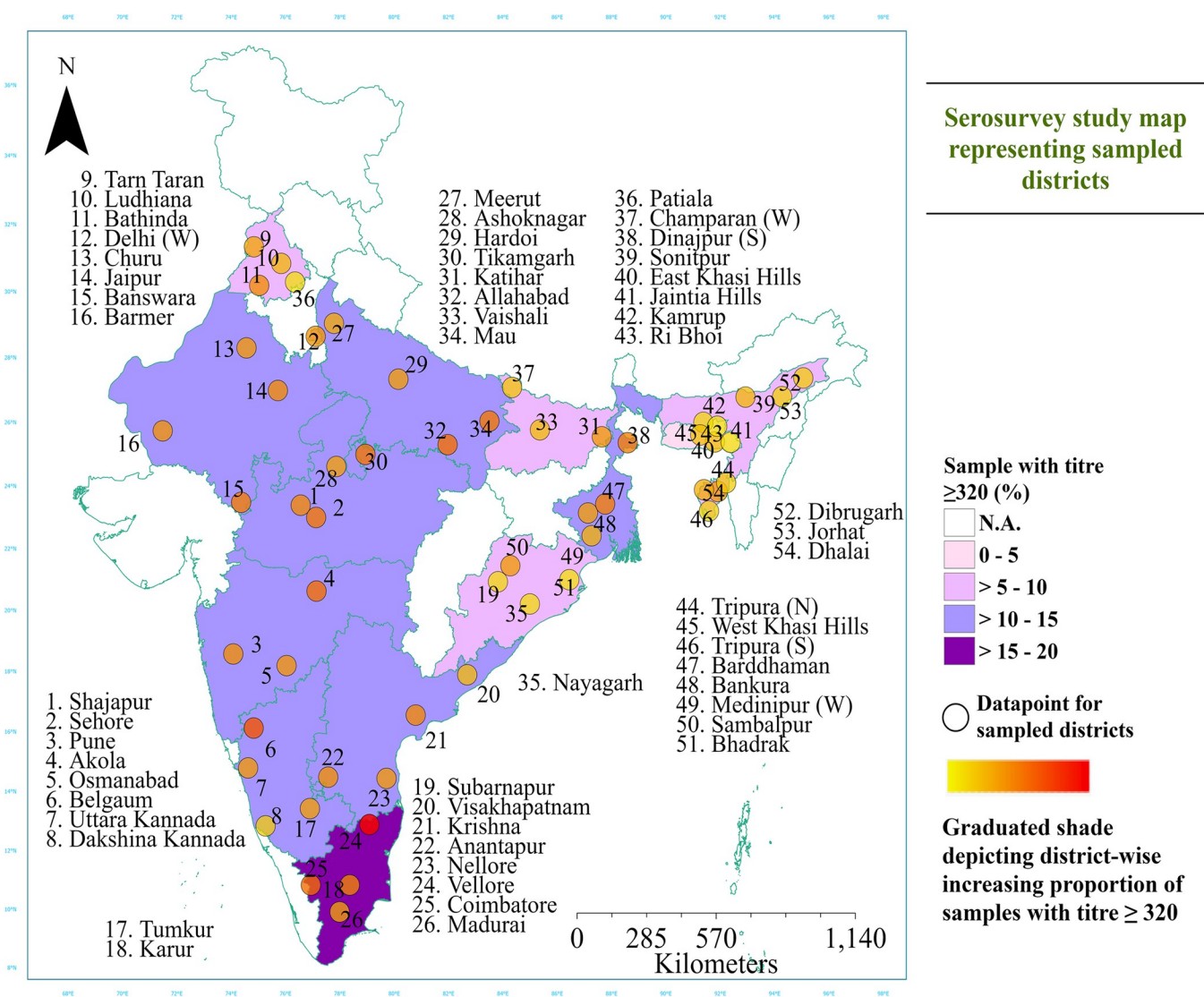

**Fig 2. Geographic representation of samples with an elevated vibriocidal titre (≥ 320) across 15 states in India.** (Note: 54 districts are labelled instead of 60 districts due to space constrains) **Note:** The base layer of the map is taken from open-source platform *https://www.indianremotesensing.com/2017/01/Download-India-shapefile-with-kashmir.html*.

Survey-4 (NFHS-4) and MPI to create a comprehensive risk map for cholera in India. The findings from the different sources were complementary, and followed similar trends, showing alignment between reported cases of cholera, serosurvey estimates, WASH coverage metrics and socio-demographic characteristics.

Of the 29,400 cases of Acute Diarrheal Diseases (ADDs) reported during confirmed cholera outbreaks, Maharashtra registered the most cases (8,040), followed by West Bengal (5,291). The top five high burden states (Maharashtra, West Bengal, Punjab, Madhya Pradesh, and Gujarat) accounted for 79.6% of all cases, in congruence with prior analyses published in 2017, which used IDSP weekly data from 2010 to 2015 to estimate the cholera burden in India. They recorded a total of 27,615 cases, with the highest caseload of 5,914 in West Bengal. This continued reporting of high burden of cholera can be traced back to evidence curated since 1996 [23]. An earlier study reporting pooled annual infection rate of cholera from 1997 to 2006

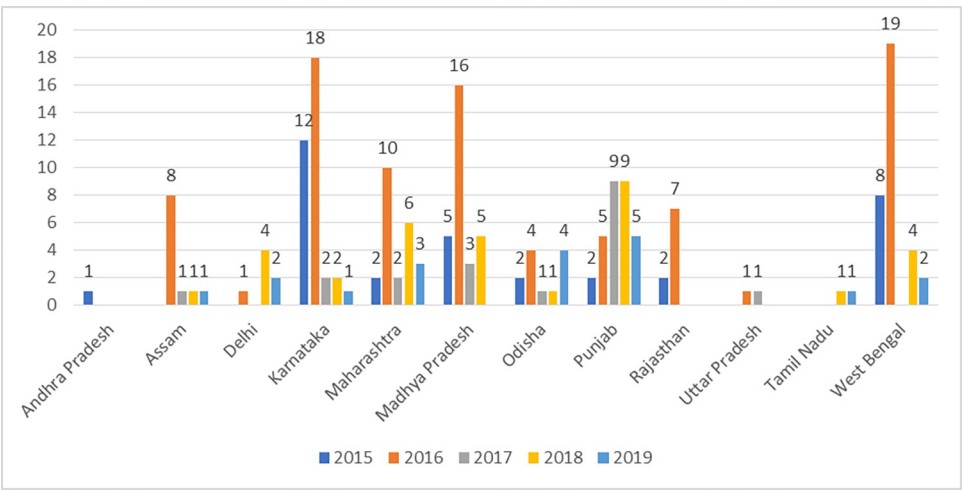

**Fig 3. Year wise reported diarrheal outbreaks from surveyed Indian states & UTs during 2015–2019 identified through literature survey.**

showed that West Bengal, Maharashtra and Odisha consistently reported higher burden of cholera [24]. In contrast to prior reporting by Ali et al., the current study estimates a marked increase in case load in Maharashtra and a decline in cases in Assam from 2015 to 2019.

Though states like Gujarat and Karnataka have reported similar numbers of cholera outbreaks when compared to West Bengal and Punjab, they report a far smaller number of cases overall, possibly indicating underreporting of cases.

The vibriocidal assay serves as a gold standard method for determining antibody development against *V. cholerae* [25]. In our study we have estimated that overall of cholera infection as 11.7% in the year before the survey (~2017). Though incomparable considering the population distribution and heterogenicity between two countries, a nationally representative serosurvey done in Bangladesh reported an adjusted annual seroincidence rate of 17·3% (95% CI 10·5–24·1) [2]. They used the term seroincidence in place of elevated vibriocidal titre ($\geq$ 320).

**Table 3. Triangulation of data between composite score & serosurvey findings alongside IDSP data during 2016.**

| State Name | *V. cholera* O1 Elevated vibriocidal titre ($\geq$ 320) | Reported diarrheal outbreaks in IDSP in 2016 | Composite Score |
|---|---|---|---|
| Tamil_Nadu | 16.7 | 0 | 88.2 |
| Madhya_Pradesh | 13.4 | 16 | 169.4 |
| Uttar_Pradesh | 12.6 | 1 | 101.3 |
| Rajasthan | 12.6 | 7 | 124.3 |
| West_Bengal | 12.1 | 19 | 168.8 |
| Karnataka | 11.5 | 18 | 170.8 |
| Maharashtra | 11.3 | 10 | 138.7 |
| Andhra_Pradesh | 10.9 | 0 | 96.1 |
| NCT_of_Delhi | 9.8 | 1 | 96.1 |
| Bihar | 8.7 | – | 109.9 |
| Punjab | 8.4 | 5 | 114.8 |
| Tripura | 7.2 | – | 67.2 |
| Odissa | 5.2 | 4 | 161.1 |
| Assam | 5.1 | 8 | 123 |
| Meghalaya | 4.4 | – | 90.7 |

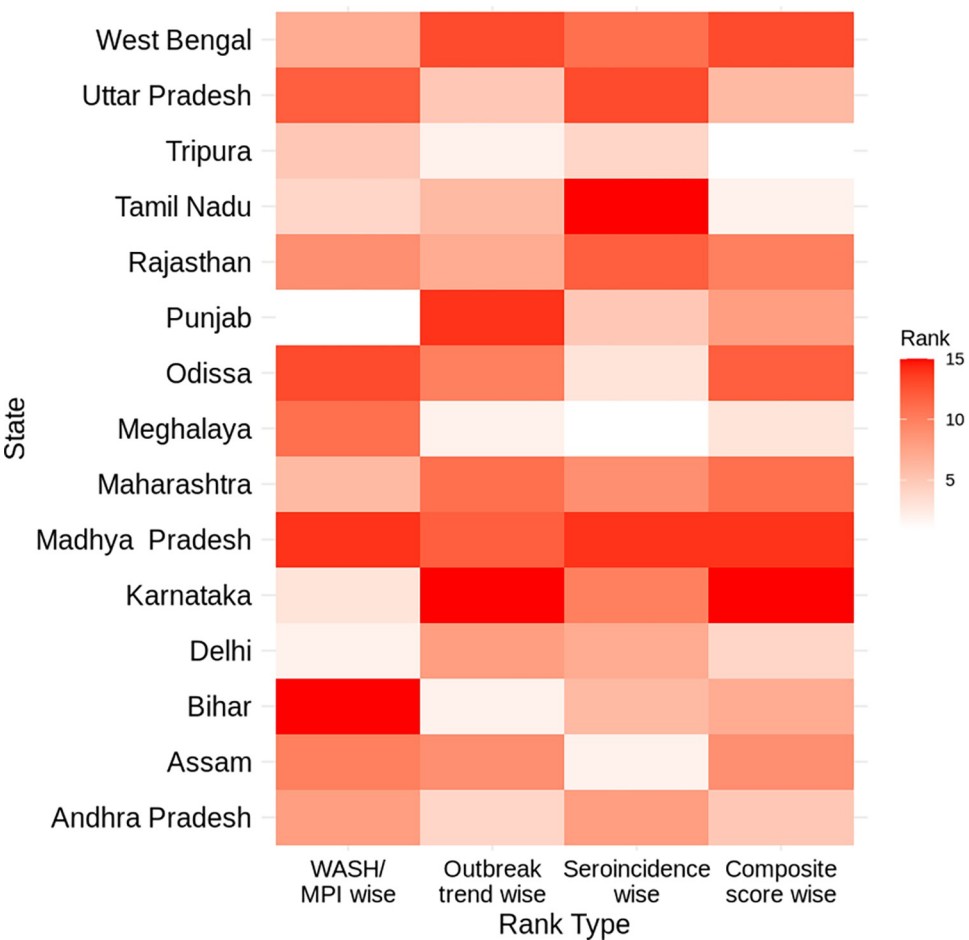

**Fig 4. Heat map comparing composite score across 15 states, to individual indicator-based ranking (WASH/ MPI, serosurvey data, outbreak trend; Note: lighter shade–low rank ➜ darker shade–more vulnerable/ higher rank).**

This finding can be compared with the 12.1% found in current study in West Bengal which is also located in Indo-Gangetic Plane similar to Bangladesh. We found that the children of age 9–17 years with an elevated vibriocidal titre ($\geq 320$) (8.6%) was significantly lower than in adults 18–45 years (13%). Previous literature also supports this finding. Comprehensive studies conducted in another cholera endemic regions of Bangladesh, assessed facility-based surveillance data from 2000 to 2021. They found that within 15 to 60 years of age group as age increase the prevalence of cholera increases from 39% to 62% within each 5-year cohort [26]. Further they reported that adjusted odds of cholera were much higher (1.93 in 15–59 years age group vs. 1.6 in 5–15 years age group) than from the reference age group of $<5$ years. Whereas, an increased attack rate of cholera is normally seen among younger and aged individuals during outbreak scenarios. Case studies on sporadic outbreaks in India by Goswami et al, Uthhappa et al. have shown attack rate as high as more than 30% in 5–14 years age group, whereas the attack rate decreases in the older age groups [27,28].

With increase in age, the exposure to cholera increases which explains this gradient. As an elevated vibriocidal titre of cholera depicts previous exposure, and the study region harbouring sporadic clusters of cholera endemicity explains the increased exposure in the higher age group in our study.

GTFCC's global roadmap to end cholera in 20 countries envisages a multi sectoral approach involving improvement in WASH, strengthening surveillance and reporting, oral cholera vaccine rollout and community engagement [29]. The absence of exact disease burden has disincentivized the process of developing a dedicated National Cholera Action Plan in India [30]. By incorporating social determinants and WASH factors, the composite vulnerability index proposed here enables policymakers to view the cholera problem in India through a wider lens, especially when it comes to identifying priority areas for targeted interventions [31].

One intriguing fact to note is that cholera was shown to be more prevalent in some of the economically stronger states, raising concerns about the composite score's validity in terms of current public health understanding of the disease. However, it must be noted that socioeconomic heterogeneity exists in all parts of India; whereas state capitals may be economically better off, serosurvey samples and cholera outbreak data are mostly collected from areas with low socioeconomic and sanitary conditions. Furthermore, as composite score comprises of multiple indicators making it more reliable as one or two indicators cannot influence the overall score alone. It is also possible that a positive detection bias is present in more affluent areas which have stronger healthcare systems as well as public health infrastructure enabling improved surveillance and reporting of cholera cases.

Even though no studies focused on index-based regional prioritization for cholera in India, researchers have developed similar vulnerability indices based on individual and environmental socioeconomic determinants for COVID-19 [31–33]. Despite differences, cholera and COVID-19 are both "society-disrupting" diseases that mostly affect low-income individuals. Both COVID-19 and cholera have the potential to change future living conditions based on hygiene and sanitation [34]. Acharya et al. published a study in 2020 that ranked states in terms of COVID-19 vulnerability [31]. Four states from the top ten most vulnerable, Madhya Pradesh, West Bengal, Orissa, and Maharashtra, align with our current study on cholera vulnerability [35]. Interestingly, Tamil Nadu though had much higher elevated vibriocidal titre among the sampled population, had much improved WASH conditions among other states in India and was ranked 14 in the composite vulnerability scale. Serosurvey as a standalone parameter does not depict cholera vulnerability in a state. Combined with outbreak trend, WASH, Multidimensional Poverty Index (MPI), it depicts the cholera vulnerability. Such score-based system, especially in lower-income countries where exact burden of disease is often unknown would greatly help in tailor-making interventions and to prioritize areas. In the above figure, it is evident that all three indicators—outbreak trend, serosurvey data & WASH/ MPI (orange, grey and blue area under the stacked lines) has greatly contributed to the composite score (yellow area).

It is evident that for a more long-term solution towards cholera prevention, the provision of Water, Sanitation and Hygiene (WASH) infrastructure is central. Access to clean water, the availability of adequate sanitation, such as basic toilets, and good hygiene practices, especially handwashing with soap, can prevent cholera outbreaks by breaking transmission routes. Thereby incorporating WASH component with a higher weightage (in MPI & separately in NFHS WASH data) in the composite vulnerability score, we though would be instrumental. No weightage was applied at the final stage while calculating the composite scores separately, as different weightage was accounted within the parent variable itself while incorporation.

## Strengths & Limitations

Our analysis has uniquely identified cholera vulnerable regions in different parts of India. It has highlighted the value of using serosurveys for complementing case-based community-centric surveillance to obtain a clear understanding of disease burden. Although the IDSP data

has been helpful for understanding cholera outbreak risk, the composite index in our analysis provides better representation of the risk of cholera by accounting for WASH and other socio-demographic risk factors. The composite risk score has helped to fine tune the risk estimates for the different states and helped expand the assessment to factors beyond case counts, which are known to be underestimated. For example, in Table 2, we see that although Tamil Nadu had the highest sero estimate, when considered in the light of the other vulnerability indicators (described in Fig 1), the composite score became 88.2. This score was lower than the scores found for states like West Bengal (168.8), Karnataka (170.8) and Madhya Pradesh (169.4) where the proportion of sampled population with an elevated vibriocidal titre ($\geq 320$) was lower. In S4 Table, we presented separate rankings using individual indicators and another without considering vibriocidal scores. These were compared to the overall composite score ranking and detailed in S4 Table. While the composite and non-vibriocidal rankings were largely similar, states with lower disease rates, like Meghalaya and Assam, performed worse in the ranking that excluded vibriocidal data. Further, the WASH/ MPI based ranking was largely dissimilar with composite score ranking (Column 6 Vs. column 9 in S4 Table) indicating WASH/ MPI data did not unduly affected the composite scores.

Our interpretation of the serosurvey data comes with some limitations. The original study was powered to estimate dengue seroprevalence at the regional- not state-level. Further, the period from June 2017 to April 2018, when the serum samples were collected for the national dengue serosurvey, reflects a point in time earlier to the present analysis. We acknowledge this limitation of our study in terms of the dynamic nature of infectious disease epidemiology, changes in risk factors, surveillance practices, and interventions over time. We used state-level sero estimates in the vulnerability index with the assumption they were representative of the entire state, and believe the use of the CVI circumvents any biases introduced by this by accounting for additional factors. The samples also mainly originated from areas with higher dengue occurrence. We recognize this as a limitation in our study as the estimates may not be extrapolated at more granular geographic levels like districts. Regarding the CVI, at this stage we have restricted our validation to content validity by sharing the tool with different experts in the fields of cholera and mathematical modelling. However, we acknowledge this as a limitation because we do not currently have a gold standard against which criteria validity can be measured. Further, this model will need validation with more robust datasets at different geographic setups to determine its validity. We would also take up a separate exercise in the future, where we will test all diarrhoeal stool samples for cholera across different regions and then compare the scores with the regional or state-wise burden.

We were also unable to test samples from those under 9 years of age. Though it is unclear how this may have influenced serosurvey estimates or relative rankings of areas, a recent study in Bangladesh found similar serosurvey estimates across age groups [36]. Further, we used vibriocidal titer as a surrogate marker for estimating recent cholera infection. Future work with additional serologic markers may improve our ability to estimate sero estimates more reliably [37]. Since the sera samples were collected for a different purpose (to estimate dengue seroincidence), only 15 states of India from six regions were included in the study, and may not necessarily reflect the states with higher burden of cholera. Analyses were done based on the region and the state instead of districts, which poses a slight challenge as national health policymaking is driven by district-level risk considerations. The current methodology may be extended, for instance by using banked samples collected through convenience sampling (i.e., blood donation drives), to find a more granular picture at the district-level. Despite these limitations, we believe the findings of the current analysis provide adequate evidence of cholera vulnerability across the country.

## Conclusion

Evidence from the current study reveals cholera infection was present in all five regions of India. Moreover, serosurveillance might represent a pragmatic approach to identify disease burden in conjunction with sentinel surveillance in high cholera burden areas alongside the existing disease outbreak surveillance system. According to the findings, states such as Karnataka, Madhya Pradesh, West Bengal, Orissa, and Maharashtra, amongst others, should be prioritised for piloting new intervention and surveillance studies to prevent and control cholera.

## Supporting information

**S1 Table. Cholera incidence using vibriocidal cut off titre of 320 in selected 60 districts of India during past one year of blood collection.**
(DOCX)

**S2 Table. Calculation of Composite Index for all 15 States involved in the study (Description of legends X1, X2... X6 has been given in Fig 1).**
(DOCX)

**S3 Table. Districts with reported outbreaks in 3 or more years identified through literature survey.**
(DOCX)

**S4 Table. Different ranking scheme against composite score ranking for different states and UTs.**
(DOCX)

**S5 Table. Sero-prevalence (%) of vibriocidal antibodies against cholera using optimized cutoff (640) \*, in different geographic regions of India, by selected socio-demographic characteristics (n = 7882).**
(DOCX)

**S1 Fig. Different administrative zones of India & the sample locations across each zone.** (The base layer of the map utilized to create this map is taken from open-source platform https://www.indianremotesensing.com/2017/01/Download-India-shapefile-with-kashmir.html).
(DOCX)

**S2 Fig. Region wise cholera incidence in different districts of India using vibriocidal cut off as 320.**
(DOCX)

## Acknowledgments

We thank the field team for their support in field operations. We thank the Director General of the Indian Council of Medical Research for his support and encouragement of the study. We acknowledge the Dengue Serosurvey team, the investigators who were part of the dengue study. We thank the entire ICMR expert panel who helped us shaping up the study and provided valuable inputs. We thank all study participants for taking part.

### Dengue Serosurvey Group

Siraj Ahmed Khan, Ramesh Reddy Allam, Pradip Barde, Bhagirathi Dwibedi, Uday Mohan, Suman Sundar Mohanty, Subarna Roy, Vivek Sagar, Deepali Savargaonkar, Babasaheb V

Tandale, Roshan Kamal Topno, C P Girish Kumar, R Sabarinathan, Sailaja Bitragunta, Gagandeep Singh Grover, P V M Lakshmi, Chandra Mauli Mishra, Provash Sadhukhan, Prakash Kumar Sahoo, S K Singh, Chander Prakash Yadav, T Karunakaran, Rajesh Kumar, G S Toteja, Nivedita Gupta, Sanjay M Mehendale.

## Author Contributions

**Conceptualization:** Suman Kanungo, Ranjan Kumar Nandy, Shanta Dutta.

**Data curation:** Rounik Talukdar, Manoj Murhekar, P Kamaraj, Muthusamy Santhosh Kumar.

**Formal analysis:** Suman Kanungo, Ranjan Kumar Nandy, Rounik Talukdar, Manoj Murhekar, Andrew S. Azman, Sonia T. Hegde, Pranab Chatterjee, Debjit Chakraborty, Madhuchhanda Das, Muthusamy Santhosh Kumar, Shanta Dutta.

**Funding acquisition:** Shanta Dutta.

**Investigation:** Suman Kanungo, Ranjan Kumar Nandy, Debjit Chakraborty.

**Methodology:** Suman Kanungo, Ranjan Kumar Nandy, Rounik Talukdar, Manoj Murhekar, Andrew S. Azman, Sonia T. Hegde, Pranab Chatterjee, Debjit Chakraborty, Madhuchhanda Das, Shanta Dutta.

**Project administration:** Suman Kanungo, Ranjan Kumar Nandy, Manoj Murhekar, Shanta Dutta.

**Resources:** Suman Kanungo, Manoj Murhekar, Andrew S. Azman, Madhuchhanda Das, Muthusamy Santhosh Kumar, Shanta Dutta.

**Software:** Suman Kanungo, Rounik Talukdar, P Kamaraj, Muthusamy Santhosh Kumar, Shanta Dutta.

**Supervision:** Suman Kanungo, Andrew S. Azman, Shanta Dutta.

**Validation:** Suman Kanungo, Ranjan Kumar Nandy, Rounik Talukdar, Manoj Murhekar, Sonia T. Hegde, Pranab Chatterjee, Debjit Chakraborty, Madhuchhanda Das, P Kamaraj, Shanta Dutta.

**Visualization:** Rounik Talukdar, Pranab Chatterjee, P Kamaraj, Muthusamy Santhosh Kumar.

**Writing – original draft:** Suman Kanungo, Ranjan Kumar Nandy, Rounik Talukdar, Manoj Murhekar, Andrew S. Azman, Debjit Chakraborty.

**Writing – review & editing:** Suman Kanungo, Ranjan Kumar Nandy, Manoj Murhekar, Andrew S. Azman, Sonia T. Hegde, Pranab Chatterjee, Debjit Chakraborty, Madhuchhanda Das, P Kamaraj, Muthusamy Santhosh Kumar, Shanta Dutta.

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
