## [Decision Letter · Decision Letter 0]

9 Apr 2024

Dear Dr. Dutta,

Thank you very much for submitting your manuscript "Understanding the risk of cholera in India through synthesis of serologic, epidemiologic, and societal determinants." for consideration at PLOS Neglected Tropical Diseases. As with all papers reviewed by the journal, your manuscript was reviewed by members of the editorial board and by several independent reviewers. In light of the reviews (below this email), we would like to invite the resubmission of a significantly-revised version that takes into account the reviewers' comments. 

We cannot make any decision about publication until we have seen the revised manuscript and your response to the reviewers' comments. Your revised manuscript is also likely to be sent to reviewers for further evaluation.

Sincerely,

Epco Hasker

Academic Editor

Stuart Blacksell

Section Editor

Reviewer's Responses to Questions

**Key Review Criteria Required for Acceptance?**

**Methods**

-Are the objectives of the study clearly articulated with a clear testable hypothesis stated?

-Is the study design appropriate to address the stated objectives?

-Is the population clearly described and appropriate for the hypothesis being tested?

-Is the sample size sufficient to ensure adequate power to address the hypothesis being tested?

-Were correct statistical analysis used to support conclusions?

-Are there concerns about ethical or regulatory requirements being met?

Reviewer #1: The objectives are clearly stated. Estimating seroincidence using vibriocidal or Ab titers now has a credible amount of data and seems a reliable indirect method. Even though this approach was developed based on data from Bangladesh, it does not seem a stretch to apply the same approach to India. However, I have some skepticism about whether the sampling was adequate to provide enough geographic granularity on burden sufficient to inform policy for interventions in such a large and heterogenous country, and also some skepticism regarding the composite vulnerability index. It is a clever approach, but the result in some geographic areas in India is not entirely logical. Specifically, the seroincidence was highest in Tamil Nadu, yet it seems primarily because of a lack of reported outbreaks the CVI resulted in Tamil Nadu having an overall ranking of 14. Therefore states or districts with very poor surveillance or reporting mechanisms could have their burden (and therefore prioritization) severely underrecognized. I wonder if the objective data of seroincidence should have more weight in the composite, along with objectively measured poverty or wash indicators?

Reviewer #2: Yes

Reviewer #3: -Are the objectives of the study clearly articulated with a clear testable hypothesis stated? YES

-Is the study design appropriate to address the stated objectives? YES

-Is the population clearly described and appropriate for the hypothesis being tested? YES

-Is the sample size sufficient to ensure adequate power to address the hypothesis being tested? YES

-Were correct statistical analysis used to support conclusions? YES

-Are there concerns about ethical or regulatory requirements being met? YES

Reviewer #4: The methods are clearly articulated, and the study design is appropriate.

**Results**

-Does the analysis presented match the analysis plan?

-Are the results clearly and completely presented?

-Are the figures (Tables, Images) of sufficient quality for clarity?

Reviewer #1: The results are clearly presented.

Reviewer #2: Yes

Reviewer #3: -Does the analysis presented match the analysis plan? YES

-Are the results clearly and completely presented? YES

-Are the figures (Tables, Images) of sufficient quality for clarity? YES

Reviewer #4: The analysis presented do match the plan. They are clearly presented

**Conclusions**

-Are the conclusions supported by the data presented?

-Are the limitations of analysis clearly described?

-Do the authors discuss how these data can be helpful to advance our understanding of the topic under study?

-Is public health relevance addressed?

Reviewer #1: The seroincidence in children 9-17y (8.6%) is significantly lower than in adults 18-45y (13%). Although this surprising finding is pointed out, no discussion or hypothesis about this observation is made. It does make me concerned about the introduction of bias in the selection of children for sampling. The methods used in the original sampling framework for Dengue seroprevalence requires some elaboration and discussion (rather than just referencing the prior publication).

Reviewer #2: (No Response)

Reviewer #3: -Are the conclusions supported by the data presented? YES

-Are the limitations of analysis clearly described? NO

-Do the authors discuss how these data can be helpful to advance our understanding of the topic under study? YES

-Is public health relevance addressed? YES

Reviewer #4: The conclusions are well presented and are based on the methods they used, but I do have major concerns about the conclusions as detailed below. The main concern relates to the over-emphasis on the WASH/SES surveys which are not validated as a true measure of cholera risk or rates. 

The public health relevance may be overstated since the authors expect the data to be useful for policy makers. However, the data is not sufficiently granular to be helpful.

**Editorial and Data Presentation Modifications?**

Reviewer #1: As stated above, I have some concerns about the CVI methodology, specifically the weighting. It might be more appropriate to have the methodology "validated" using a more robust set of data with granular seroincidence and/or intense surveillance to verify that it predicts risk well, before applying it to an entire country as large and heterogenous as India. Could the CVI be validated in Bangladesh or a portion of India where the seroincidence coverage is high, along with strong data on Wash indicators, and where reporting systems are know to be strong? In particular it would be important to know if the reported cases/outbreak component should be considered in the CVI.

Some further discussion about the surprisingly low seroincidence in 9-17y seems warranted.

Reviewer #2: Accept

Reviewer #3: (No Response)

Reviewer #4: See comments below.

**Summary and General Comments**

Reviewer #1: Overall this is a very ambitious work attempting to develop a methodology to identify and prioritize cholera risk areas in India. These methods could be applied to other countries as well. However, I am not certain the methods (specifically CVI) have been sufficiently validated to apply at this large of a scale. Once validated and refined, this type of composite analysis (seroincidence combined with poverty/wash indices) could be very helpful to countries in developing intervention priorities. I have indicated a "major revision" because a validated composite analysis would be extremely useful and one that gives inaccurate results very harmful.

Reviewer #2: No additional comments

Reviewer #3: General comment

This is a well-written study on cholera in India by Kanungo et al. The authors analyzed data on cholera by testing 7,882 stored sera samples collected during 2017-18 from individuals aged 9-45 years. They assessed vibriocidal antibodies to Vibrio cholerae O1 using a cut-off titre of ≥320, defining seroincidence. Additionally, they extracted data from the 2015-19 Integrated Disease Surveillance Programme and published reports.

The authors found that overall, 11.7% (CI: 10.4-13.3%) of the sampled population had seroincident infection. Furthermore, they established that between 2015-2019, the IDSP reported 29,400 cases of cholera across India. These findings are interesting and have the potential to attract actors involved in cholera prevention and research. However, there are several issues that remain unclear in the current version of the manuscript.

Major comments

1. Title, “Understanding the risk of cholera in India through synthesis of serologic, epidemiologic, and societal determinants”, This title, as it stands, is open to speculation and does not fully reflect the content of the manuscript. For instance, the data and reports used to generate the manuscript were from the period 2015-2019. I think that the findings of this study would differ from those of a similar study with the same title but using data from 2020-2023 or reports such as the National Family and Health Survey 2020-2021, https://main.mohfw.gov.in/sites/default/files/NFHS-5_Phase-II_0.pdf. Hence, the authors should revise the title to ensure that it reflect the manuscript content.

2. Lines 89-91, “ We generated vibriocidal data using serum samples (n = 7882) collected for the national dengue serosurvey, conducted by ICMR National Institute of Epidemiology (ICMR-NIE), Chennai, India from June 2017 to April 2018.12”. The period when the sera samples were collected is far back, ie 5-6 years from 2023. Hence, it is possible that the risk factors associated with cholera at that time were addressed. This is reflected in the fact that National Family and Health Survey (NFHS-4, 2015-2016) data was found inadequate for planning by the Indian authorities who subsequently invested in NFHS-5 (2020-2021), https://main.mohfw.gov.in/sites/default/files/NFHS-5_Phase-II_0.pdf. Therefore, the authors need to state this long period past the present time as an important consideration in interpretation of the study findings. 

3. Lines 115-116, “at -80°C till accessed for vibriocidal assays. Sera from children aged 5-8 years could not be analysed due to inadequate sample volume”. This is an important statement. The omission of children aged 5-8 years due to inadequate sample volume is an issue that may attract attention from actors involved in child protection and rights, such as UNICEF. Therefore, it would be useful if the authors explicitly stated the volume of sera that was deemed adequate. Additionally, they should include in the exclusion criteria a statement indicating that all participants with sera below the stated volume were excluded. Such a statement, in addition to improving the clarity of the information shared, has the potential to encourage researchers to conduct studies that can generate useful and reliable data from small volumes of sera specimens

Other comments 

4. Lines 88-93, “The study had been…. ” The authors use some abbreviation such as ICMR, for the first time without defining them. The authors need to define all such abbreviations to improve on the manuscript readability. 

5. Lines 86-108, the authors frequently use the term 15 states yet India has 28 states and 8 territories, https://en.wikipedia.org/wiki/List_of_Indian_state_days . It would be useful if the information on major administrative units of India was included in this section to enable the readers contextualize the message in the manuscript. 

6. Lines 110-111, “In this study, we estimated the seroincidence of cholera among individuals aged 9 to 45 years in the year preceding sample collection in 2017”. This statement could be better if the year was stated e.g in 2016 if that is what the authors meant.

Reviewer #4: This paper represents an attempt to assess the cholera disease burden by using a new scoring method which includes data vibriocidal titers from serum samples collected earlier with case and outbreak numbers, and from WASH and SES surveys. By adding the numbers from each of these variables, they attempt to rank different states in terms of their cholera burden. They emphasize that the routine surveillance of cholera is not sufficient nor accurate because the cholera infections are often asymptomatic and because the surveillance systems in may not be uniformly administered between states and districts. They believe that a composite score may help to correct these deficiencies. They also hope that this type of method will help policy makers focus control interventions.

This paper does provide such a scoring system, but it also has several constraints that need to be highlighted.

1. The data at the state level is not sufficiently granular to help focus control efforts. One would need to have data at the district, or better, a subdistrict, to be really useful. However, the type of WASH and SES is not generally available at this granular levels.

2. It seems excessive weight is given to the WASH/SES data in the composite score. While most of the paper stresses the serology and the surveillance numbers, in fact the composite score seems to be driven by the WASH and SES surveys. In addition to the question of whether this places too much weight to these surveys, these indicators may, in fact, not be a good indicator of disease burden or risk. Data from surveys are taken from a random sample of subjects in the defined area, but cholera occurs in select group of households which are generally outlier households with exceptionally poor WASH/SES status, and are not representative of the data from the random sample of households. 

An example of this overweighting is from the Figure 1 and Table III of the Annex. The example in Figure 1 shows that data from the WASH and SES surveys represent 83% of the total score while the biologic measures contribute only 17% to the total score. In Table III, although I did not do similar calculation, it seems that the WASH and SES are the predominant drivers of the score. 

3. I realize that these titers (vibriocidal titer of 320) were evaluated in Bangladesh as indicative of annual “incidence” rates. It is notable however, that the titers in the older age strata is somewhat higher than the younger age strata. Does this really represent higher incidence in this age group or is it more likely that the increased rate of high titers may be due to longer persistence of higher titers in the older group who may have been exposed earlier and have a longer-lasting (booster) response. Have the authors carried out an analysis of the Bangladesh serum data in an attempt to determine if higher titers persist longer in older age groups? Because of this concern about the term incidence, it would seem more accurate to replace the term "incidence" with the term “elevated titers.” One can still suggest that the rate of elevated titers relates to incidence, but I believe this change in wording is more accurate.

In addition, here are few detailed comments.

Line 71 is a mis-statement of the GTFCC goals. This should be a 90 % reduction in deaths and elimination in 20 countries.

Line 94 should include references for MPI and National Family Health Survey-4.

Line 107 correct wording to indicate that these are serum samples from 12,300 individuals. I assume these were anonymous. I did not find the word anonymous in the manuscript. 

Line 165 mentions O139. Were any cases associated with serotype O139? If not, this should be mentioned. 

Line 174 mentions that many (around 80%) of cholera infections are asymptomatic or at least do not seek medical attention, but these estimates are primarily from family studies which were associated with symptomatic cases. Whether ongoing silent transmission, in the absence of outbreaks, is not so clear. Are the authors suggesting that silent transmission occurs or is likely or is it more likely that these additional asymptomatic cases are more likely to occur when other subjects have detectable disease. 

Line 237. I note that West Bengal is not included in the list of states with cholera each year. This seems strange since I expected cholera to occur in Kolata every year. How to explain this? Since the authors are based in Kolkata, do they believe this to be true? 

Line 243 discussed cases and outbreaks. Later in the tables, these are differentiated, and it seems this needs to be clarified to clearly define “cholera cases” and “outbreaks.” If there were isolated cholera cases reported (not part of an outbreak), were these confirmed?

Numbers of cases: The authors may want to clarify that the numbers of cases in this paper do not correlate with the numbers reported to WHO. These numbers reported to WHO were as follows: 2015 (889), 2016 (841), 2017 (385), 2018 (697), 2019 (0 cases). I do not know how to explain these differences. 

Table 2 does not show a relation between the number of cases or the number of outbreaks with the serum titers. Do the authors have a comment about this lack of correlation?

Annex Table 1. There is no need to have two columns for the number of sera with titers above and below 320. This can be a single column and the percentage. 

Table V The title refers to IgG titers which must be a typo.

PLOS authors have the option to publish the peer review history of their article (what does this mean?). If published, this will include your full peer review and any attached files.

Reviewer #1: No

Reviewer #2: Yes: Denny John

Reviewer #3: No

Reviewer #4: No
---

## [Decision Letter · Decision Letter 1]

10 Jul 2024

Dear Dr. Dutta,

Thank you very much for submitting your manuscript "Analysis of Cholera Risk in India: Insights from 2017-18 Serosurvey Data Integrated with Epidemiologic data and Societal Determinants from 2015-2019" for consideration at PLOS Neglected Tropical Diseases. As with all papers reviewed by the journal, your manuscript was reviewed by members of the editorial board and by several independent reviewers. The reviewers appreciated the attention to an important topic. Based on the reviews, we are likely to accept this manuscript for publication, providing that you modify the manuscript according to the review recommendations. 

Sincerely,

Epco Hasker

Academic Editor

Stuart Blacksell

Section Editor

Reviewer's Responses to Questions

**Key Review Criteria Required for Acceptance?**

**Methods**

-Are the objectives of the study clearly articulated with a clear testable hypothesis stated?

-Is the study design appropriate to address the stated objectives?

-Is the population clearly described and appropriate for the hypothesis being tested?

-Is the sample size sufficient to ensure adequate power to address the hypothesis being tested?

-Were correct statistical analysis used to support conclusions?

-Are there concerns about ethical or regulatory requirements being met?

Reviewer #3: -Are the objectives of the study clearly articulated with a clear testable hypothesis stated? YES

-Is the study design appropriate to address the stated objectives? YES

-Is the population clearly described and appropriate for the hypothesis being tested? YES

-Is the sample size sufficient to ensure adequate power to address the hypothesis being tested? YES

-Were correct statistical analysis used to support conclusions? YES

-Are there concerns about ethical or regulatory requirements being met? YES

Reviewer #4: this is acceptable

**Results**

-Does the analysis presented match the analysis plan?

-Are the results clearly and completely presented?

-Are the figures (Tables, Images) of sufficient quality for clarity?

Reviewer #3: Does the analysis presented match the analysis plan? YES

-Are the results clearly and completely presented? YES

-Are the figures (Tables, Images) of sufficient quality for clarity? YES

Reviewer #4: This is acceptable

**Conclusions**

-Are the conclusions supported by the data presented?

-Are the limitations of analysis clearly described?

-Do the authors discuss how these data can be helpful to advance our understanding of the topic under study?

-Is public health relevance addressed?

Reviewer #3: -Are the conclusions supported by the data presented? YES

-Are the limitations of analysis clearly described? YES

-Do the authors discuss how these data can be helpful to advance our understanding of the topic under study? YES

-Is public health relevance addressed? YES

Reviewer #4: I do not like the model used by the authors since I continue to feel that too much weight it given to the WASH and SES data even though most of the manuscript is taken up by the description of the methods and results of the vibriocidal antibody testing. Since vibriocidal results contribute such a small amount to the score, it seems the overall scores might have yielded the same ranking even without including the vibriocidal results. On line 409 the authors explain "sharing the tool to different experts in the field of cholera and mathematical modelling, it is unanimously agreed as the best possible index with the limited data availability." I do not think this is an adequate justification. I believe there should be some additional wording to indicate that this model will need to be evaluated by others to determine its validity (or something to this effect) to indicate this is a limitation.

**Editorial and Data Presentation Modifications?**

Reviewer #3: (No Response)

Reviewer #4: Line 426 typo “convenience”

**Summary and General Comments**

Reviewer #3: I have essential revision required.

Results lines 242-245, “(Annexure: Table II). A total of 268 diarrheal outbreaks were reported to the IDSP, five states (West Bengal, Karnataka, Punjab, Madhya Pradesh, and Gujarat) accounted for 60% of these outbreaks. <Figure 2: Year wise reported diarrheal outbreaks from surveyed Indian states & UTs during 2015 - 2019 identified through literature survey>” and 

Discussion section Line 263- 266, “Of the 29,400 cases of Acute Diarrheal Diseases (ADDs) reported during confirmed cholera outbreaks, Maharashtra registered the most cases (8,040), followed by West Bengal (5,291). The top five high burden states (Maharashtra, West Bengal, Punjab, Madhya Pradesh, and Gujarat) accounted for 79.6% of all cases, in congruence with prior analyses published in 2017”. 

The authors rightly highlight the significance of acute watery diarrhea (AWD) as a major public health concern. However, I note a discrepancy in their recommendations, which predominantly focus on cholera while neglecting AWD. To strengthen the manuscript's impact and comprehensiveness, I suggest the authors provide equally robust recommendations for AWD, acknowledging its equal importance in the context of waterborne diseases. By doing so, the authors can ensure a more balanced and effective approach to addressing the burden of diarrheal diseases.

Reviewer #4: no other comments

PLOS authors have the option to publish the peer review history of their article (what does this mean?). If published, this will include your full peer review and any attached files.

Reviewer #3: Yes: Godfrey Bwire, MBChB, MPH, PhD

Reviewer #4: No

Figure Files:

Data Requirements:

Reproducibility:

References

---

## [Editor Report · Decision Letter 2]

12 Aug 2024

Dear Dr. Dutta,

We are pleased to inform you that your manuscript 'Analysis of Cholera Risk in India: Insights from 2017-18 Serosurvey Data Integrated with Epidemiologic data and Societal Determinants from 2015-2019' has been provisionally accepted for publication in PLOS Neglected Tropical Diseases.

Best regards,

Stuart D. Blacksell

Section Editor

---

## [Editor Report · Acceptance letter]

27 Aug 2024

Dear Dr. Dutta,

We are delighted to inform you that your manuscript, "Analysis of Cholera Risk in India: Insights from 2017-18 Serosurvey Data Integrated with Epidemiologic data and Societal Determinants from 2015-2019," has been formally accepted for publication in PLOS Neglected Tropical Diseases.

Best regards,

Shaden Kamhawi

co-Editor-in-Chief

Paul Brindley

co-Editor-in-Chief
